# Synthesis and Performance Evaluation of a Novel Heat and Salt-Resistant Gel Plugging Agent

**DOI:** 10.3390/polym14183894

**Published:** 2022-09-17

**Authors:** Xuejiao Li, Meilong Fu, Jie Liu, Qi Xiao, Wenhao Tang, Guike Yang

**Affiliations:** 1Hubei Key Laboratory for Processing and Application of Catalytic Materials, College of Chemistry and Chemical Engineering, Huanggang Normal University, Huanggang 480000, China; 2College of Petroleum Engineering, Yangtze University, Wuhan 430000, China; 3School of Earth Science, Northeast Petroleum University, Daqing 163318, China

**Keywords:** high-temperature and high-salinity reservoir, naphthol gel plugging agent, performance evaluation

## Abstract

Tahe oil field is a typical fissure cave carbonate reservoir with a temperature of up to 120~140 °C and a total salinity of formation water of (20~25) × 10^4^ mg/L. In this paper, AM/AMPS was selected as the polymer, 1, 5-dihydroxy naphthol as the cross-linking agent, and polypropylene fiber as the system stabilizer to synthesize a novel gel plugging agent independently; the gel has good thermal stability at a high temperature of 130 °C and increased salinity of 20 × 10^4^ mg/L, and has a dense and relatively stable three-dimensional network structure under a scanning electron microscope. The performance evaluation of the gel plugging agent indicated that: the gel dehydration rate increased gradually with the increase in temperature and salinity, making it suitable for reservoirs with temperatures below 140 °C and formation water salinity below 250,000 mg/L; the viscosity of the gel bulk was 125.3 mPa∙s, the post-gelatinizing viscosity was 42,800 mPa∙s, and the gelatinizing time at 120 °C, 130 °C and 140 °C was 10–20 h, 8–18 h, and 7–16 h, respectively.

## 1. Introduction

Around 60% of the world’s total oil and gas production comes from carbonate reservoirs, and 40% of the entire oil and gas reserves are in carbonate reservoirs [1,2]. Tahe oil field is a typical fracture cave carbonate reservoir with a buried depth of 5300–7000 m. The formation temperature is between 120 °C and 140 °C. The total salinity of formation water is up to (20–25) × 10^5^ mg/L. The reservoir is characterized by ultra-depth, high temperature, high salinity, substantial heterogeneity, etc. Its complex geological conditions and mining difficulties are infrequent in the world. At present, the main challenges encountered in the extraction process of the Tahe oil field are rapid increase in water content of oil wells, immediate production reduction of oil wells, and overall deviation of the water plugging effect; therefore, it is genuinely urgent to slow down the increasing rate of water content after water was seen in the oil wells [3].

Most gel plugging agents have a colloidal and reticular structure, and water can fill the gel lattice after gelatinizing, with favorable viscoelasticity [4]. Gels mainly block high-permeability reservoirs relying on physical clogging, trapping, and adsorption and are suitable for plugging fractured reservoir bodies. However, current polymer gels have poor stability in high-temperature and high-salinity reservoirs, and ordinary HPAM is prone to degradation at high temperatures, resulting in gel dehydration and shrinkage [4], which is not suitable for the high-temperature and high-salinity reservoirs in the Tahe oil field. Compared with HPAM, acrylamide copolymers, e.g., acrylamide/tert-butyl acrylate copolymer (PAtBA), acrylamide/dimethyl diallyl ammonium chloride copolymer (AM/DMDAAC), and acrylamide/2-acrylamide-2-methyl propane sulfonic acid copolymer (AM/AMPS), have better temperature and salinity resistance, as the monomers in these copolymers have better temperature and salinity resistance than carboxyl groups. The large volume of monomers increases the steric hindrance of molecules, thereby inhibiting the hydrolysis of amide groups and improving the temperature and salinity resistance of the polymers [5,6]. Therefore, it is still significantly difficult to discover novel temperature and salinity-resistant cross-linkers, screen temperature and salinity-resistant polymers, and develop and synthesize gel plugging agents suitable for high-temperature and high-salinity reservoirs.

In this paper, a novel temperature and salinity-resistant gel plugging agent was independently developed, with good thermal stability at a high temperature of 130 °C and high salinity of 20 × 10^4^ mg/L. In this work, AM/AMPS was selected as the polymer; an AM/AMPS polymer was polymerized by acrylamide (AM) and AMPS monomer. After the polymerization of the acrylamide and AMPS monomer, the steric hindrance of the polymer molecule significantly increased, effectively inhibiting the hydrolysis of amide groups, empowering strong stability to the polymer, and considerably improving its heat resistance and salt tolerance. 1,5-dihydroxy naphthol was selected as the main cross-linking agent. 1,5-dihydroxy naphthol has one naphthalene ring, the two benzene rings each contained a hydroxyl group naphthalene ring, the double benzene rings were activated, and more chemical reaction sites were generated, quickly obtaining a net-like gel structure with the skeleton extending in four directions. In addition, a gel performance evaluation was performed on the effects of temperature and salinity on gel dehydration rate, gel viscosity, and gelatinizing time. This paper provides favorable technical support for the field application of the gel plugging agent in the Tahe oil field.

## 2. Experimental Section

### 2.1. Experimental Supplies

The polymers used in this paper mainly include AM/AMPS, with a solid content of >88%, AMPS content of >50%, and viscosity average molecular weight of 5 million; AM/DMDAAC, polymer AP915, with viscosity average molecular weight of 5 million and ionic strength of 30%; HPAM, with a molecular weight of 8 million and solid content of >88%. Polymers were purchased from Shandong Baomo Biochemical Co., Ltd (Dongying, China).

Other experimental drugs include resorcinol, hydroquinone, catechol, polyethyleneimine, polyethylene polyamine, α-naphthol, 1,5-dihydroxy naphthol, methanal, thiourea, sodium hydroxide, and polypropylene fiber. All drugs were analytically pure and provided by Shanghai Macklin Biochemical Technology Co., Ltd.

Experimental water: simulated formation water prepared according to the ion composition of the produced water in the Tahe oil field, with a total salinity of up to 220,000 mg/L and a calcium and magnesium ion content of up to 12,000 mg/L.

Experimental oil: dehydrated and degassed crude oil with the ground viscosity of 55 mPa·s.

### 2.2. Experimental Apparatus

DV2T viscosimeter (Brookfield, IL, USA); ZA220R4 precision electronic balance (maximum range 220 g, readability 0.1 mg), Shanghai Zanwei Weighing Apparatus Co., Ltd. (Shanghai, China); 85-2 thermostatic magnetic stirrer, Changzhou Longhe Instrument Manufacturing Co., Ltd. (Changzhou, China); precision aging tank (6-bore screws, 500 mL volume), Jiangsu Lianyou Scientific Research Instrument Co., Ltd. (Nantong, China); EM-30AX scanning electron microscope (SEM), COXEM, (Daejeon, Korea); large constant temperature oven (0–250 °C), Jiangsu Lianyou Scientific Research Instrument Co., Ltd.

### 2.3. Experimental Method

#### 2.3.1. Determination of Gel Dehydration Rate

The gluing solution was arranged in a plurality of ampoules, and after being glued, it was taken out from the thermostat. After opening the ampoule bottle, the gel was weighed. The dehydration rate was defined as the ratio of the obtained weight to the initial gel weight.

#### 2.3.2. Determination of Gel Strength

The gel strength was qualitatively determined by Sydansk Gel Strength Codes [7], as shown in Table 1.

#### 2.3.3. Polymer Viscosity Retention Test

Prepare the polymer mother liquor, measure the polymer viscosity μ_0_ at room temperature with a viscometer, put the polymer in an ampoule, and seal it. Then, put it in a specific high-temperature oven for some time, and take out the polymer to determine its viscosity μ_1_. The calculation formula follows polymer viscosity retention = (μ_1_/μ_0_) × 100%.

#### 2.3.4. Gel Formulation

First, the polymer is prepared into a mother solution according to the set concentration for use, and the simulated formation water is prepared according to the ion composition of the fake formation water; rather, the pH value of the system was 8, and the system is stirred evenly with a magnetic stirrer; then an appropriate amount of gel stock solution is added to a specific ampoule bottle, and the mouth of the ampoule bottle is fired and sealed; finally, the ampoule bottle is put into the aging tank, and the aging tank is placed in a 130 °C incubator to observe the gelation of the gel.

## 3. Results and Discussion

### 3.1. Synthesis of a Novel Heat and Salt-Resistant Gel Plugging Agent

#### 3.1.1. Determination of Gel Polymers

In this paper, three acrylamide copolymers of AM/AMPS, AM/DMDAAC, and polymer AP915, as well as common HPAM [8], were screened to investigate the thermal stability of various polymers at high temperature and high salinity (simulated formation water, 130 °C), and the experimental results are shown in Table 2.

According to the experimental data in Table 2, when treated at high temperature and high salinity for 6 h, 12 h, and 24 h, the viscosity retention of polymer AM/AMPS was 89%, 78.4%, and 68.2%, respectively, and the viscosity retention peaked. By observing this phenomenon, AM/AMPS was still relatively transparent after heating at high temperatures, with no significant change in appearance and shape, and the thermal stability was the best. At the same time, the viscosity retention of the other three polymers decreased to below 10% 24 h later, and all of them turned turbid to varying degrees after heating at high temperatures since white precipitates or flocs, produced by the reaction of carboxyl groups with Ca^2+^ in such polymers, were formed in these solutions, indicating that the thermal stability of the three polymers was relatively poor. Upon analysis, it was believed that AM/AMPS polymer was polymerized by the acrylamide (AM) monomer and AMPS monomer; for AMPS monomer, the highly stable carbon chain was its main chain, and anionic solid hydration groups, i.e., methyl propyl sulfonate groups, were introduced into its molecule, which not only improved the hydrophilicity, but also greatly improved the salinity resistance of the AMPS monomer thanks to the hyposensitivity of the propane sulfonic acid groups to external cations; after the polymerization of the acrylamide and AMPS monomer, the steric hindrance of polymer molecule significantly increased, effectively inhibiting the hydrolysis of amide groups, empowering strong stability to the polymer, and considerably improving its heat resistance and salt tolerance, which laid a solid foundation for the subsequent preparation of heat-resistant and salt-tolerant gels.

Therefore, in this paper, AM/AMPS was selected as one of the essential raw materials for the synthesis of novel gels.

#### 3.1.2. Determination of Gel Cross-Linkers

Seven common heat-resistant and salt-tolerant cross-linkers were screened for cross-linking experiments [9], among which a-naphthol and 1,5-dihydroxy naphthol were used as cross-linkers for the first time in the synthesis of high-temperature gels, which belonged to the original synthesis idea. The experimental results are shown in Table 3.

According to Table 3, the dehydration rate of the two naphthol cross-linkers was the lowest, 7.6% and 6.8%, respectively. This is mainly because the naphthols are phenolics containing rigid group naphthyl; compared with conventional naphthols, the phenolic resin cross-linker formed by such phenolics has better heat resistance, thus making the gelatinized gel more stable at high temperatures.

In the experiment, gel 1,5-dihydroxy naphthol showed higher stability because its double benzene rings each contained a hydroxyl group, and it had one more naphthalene ring than α-naphthol; as a result, double benzene rings were activated, and more chemical reaction sites were generated, so that it was more likely to form free hydroxymethyl groups in the presence of excessive methanal to better cross-link with the polymer AMPS and quickly obtain a net-like gel structure with the skeleton extending in four directions, as shown in Figure 1.

Therefore, 1,5-dihydroxy naphthol was selected as the novel heat- and salinity-resistant gel plugging agent’s leading cross-linking agent.

#### 3.1.3. Screening and Dosage of Stabilizer

To improve the long-term stability of the system under high-temperature and salinity conditions, seven different types of materials were screened and added to the system as stabilizers to investigate the strength and dehydration rate of the system at 30 days using the gel prepared with Tahe’s simulated formation water at 130 °C.

According to Table 4, adding fiber stabilizers to the naphthol gel system could effectively improve the long-term stability of the system, among which polypropylene fiber (6 mm) had the best effect. The gel strength was still graded H after 30 days of aging under high temperature and salinity. The dehydration rate was only 3%, therefore, polypropylene fiber was selected as the system stabilizer.

Polypropylene fiber could be evenly dispersed when added into the naphthol gel system to improve the gel network structure’s compactness and the gel’s water retention capacity. The additive amount of polypropylene fiber was 0.05%, 0.1%, 0.2%, 0.3%, 0.4%, 0.5%, and 0.6%, and other conditions remained unchanged. The gel was prepared with simulated formation water from the Tahe River, and the system was placed at 130 °C for a cross-linking reaction for five days. The experimental results are shown in Figure 2.

It was figured out from Figure 2 that within the polypropylene fiber-specific concentration range, the gelatinizing strength of the system gradually increased with the increase in content; when the fiber content reached 0.5%, the gelatinizing power of the system reached grade H, and the dehydration rate in 5 days was only 4%; when the fiber content continued to increase, the gelatinizing strength and dehydration rate of the system did not change much, and 0.5% was the optimal concentration of stabilizer polypropylene fiber.

Polypropylene fiber was a safe, non-toxic, acid- and alkali-resistant, diamagnetic, rust-proof, heat-resistant, and salt-tolerant synthetic fiber, which did not participate in the chemical cross-linking reaction in the entire gel system but only strengthened the three-dimensional network framework of the naphthol gel system. Polypropylene fibers mainly form hydrogen bonds with the molecular chains of the gel. When the naphthol gel stock solution was dynamically injected, hydrogen bonds were generally broken, which means that the viscosity of the stock remained unchanged before and after adding fibers into the gel. When the naphthol gel plugging agent reached the plugging formation, the gel stock solution was in a static state and began to cross-link into a network structure; meanwhile, the fibers could bond with the polymer chain by hydrogen bonds, which increased the critical points of the gel system, improved the tensile and compressive strength of the system, and greatly improved the stability of the gel system structure, as shown in Figure 3. In addition, polypropylene fibers, by their strong hydrophilicity, improved the gel’s ability to bind water molecules in the spatial grid structure and the water retention performance of the gel system.

#### 3.1.4. Manifestations of Gel Microstructure

The naphthol gel was dark brown after gelatinizing, with high gel strength and good viscoelasticity, as shown in Figure 4

The structural components of the optimal naphthol gel plugging agent studied were simply characterized and analyzed by FT-IR analysis. The specific results are shown in Figure 5.

Both 2926.15 cm^−1^ and 2864.07 cm^−1^ are characteristic peaks of the bishydroxynaphthalene ring; 1629.46 cm^−1^ is the absorption peak of primary amide, and the absorption type is stretching vibration [10]; 1203.89 cm^−1^ is the phenolic (C-O) absorption peak (1000–1260), the absorption type is stretching vibration; 871.11 cm^−1^ is the absorption peak of the 1,2,3,4-position four-membered benzene ring substitution, and the absorption type is out-of-plane bending [11].

The typical chromium gel and the novel naphthol gel were analyzed by SEM [12] and the microstructure of the latter was analyzed and compared, as shown in Figure 6.

As can be seen from Figure 5, compared with standard chrome gel, the novel naphthol gel was densely net-like, with a dense skeleton and a thick trunk (closely connected with branches); the meshes in the net overlapped each other and varied in size, indicating that the naphthol gel formed a three-dimensional net-like structure. Such structure was mainly obtained by the cross-linking of long-chain macromolecules, which were relatively stable and could better confine water molecules in the spatial grid to form a colloid with strong viscoelasticity and good stability.

### 3.2. Performance Evaluation of a Novel Heat and Salt-Resistant Gel Plugging Agent

#### 3.2.1. Effect of Temperature on the Gel Dehydration Rate

A novel gel plugging agent was prepared under five high-temperature conditions of 100 °C, 110 °C, 120 °C, 130 °C, 140 °C, and 150 °C, and the aging time of three days, five days, eight days, and ten days was set. The effect of high-temperature conditions on the gel dehydration rate was investigated and the experimental results are shown in Figure 7.

According to Figure 7, the novel naphthol gel plugging agent had excellent resistance at elevated temperatures, especially within 100–140 °C; no gel breaking was found in 10 days, the strength reached grade H, and the dehydration rate was less than 3.6%. The main reasons were as follows: the naphthol gel cross-linker contained naphthalene rings with rigid groups, which had solid thermal stability. The polymer AM/AMPS had a relatively large monomer volume, which increased the steric hindrance of molecules, inhibited the hydrolysis of amide groups, and improved the stability of long-chain macromolecules of the polymer at high temperatures. In the gel, the cross-linker and polymer were mainly cross-linked in the form of covalent bonds, forming a three-dimensional network structure whose skeleton developed in four directions, and this structure was relatively stable; Polypropylene fiber, a stabilizer, was added to the system, which could form hydrogen bonds with the molecular gel chain to stabilize the network structure further and effectively improve the water retention capacity of the gel [13].

The gel dehydration rate increased gradually with the increase in temperature, especially when the temperature reached 150 °C, the gel strength began to decline, and the gel dehydration rate increased significantly. The dehydration rate in 10 days reached 9.8%. The gel aged at 130 °C and 150 °C for ten days was analyzed by SEM, as shown in Figure 8.

According to (a) and (b) in Figure 8, by comparing the gel structures at the exact multiple, it could be seen that at 130 °C, the gel structure was stable, the network skeleton overlapped intensively, and the meshes were uniformly distributed; at 150 °C, the gel network structure was sparse, the mesh spacing was relatively large, and the traces of fracture were observed at the edges of the network structure. The SEM multiple was further increased and upon the comparison with Figure 8c,d, it could be seen that at 130 °C, the main gel chain was strong and stable, and various branches were closely connected with the main chain; while at 150 °C, it was obvious that the side chains of different branches in the structure were utterly broken, even part of the main chain was broken [13,14]. Therefore, the high temperature could destroy the gel network structure and reduce its capacity for confining water molecules and the gel dehydration rate increased accordingly.

In conclusion, the novel naphthol gel was suitable for reservoirs below 140 °C. After aging at 100–140 °C for ten days, the gel dehydration rate changed little, the gel strength reached grade H, the dehydration rate was less than 3.6%, and the heat resistance was favorable. Considering that the formation temperature of the Tahe oil field is between 120 °C and 140 °C, the naphthol gel can meet the needs of high-temperature reservoirs in the Tahe oil field.

#### 3.2.2. Effect of Salinity on the Gel Dehydration Rate

The salinity of simulated formation water in the Tahe Oil field was 220,000 mg/L. The simulated formation water was diluted by 20%, 40%, 60%, and 80% with distilled water and then 3% NaCl was added to prepare water samples with higher salinity to investigate the salt tolerance of the naphthol gel under different salinities. The aging time of each water sample was three days, five days, eight days, and ten days, and the experimental temperature was 130 °C. The experimental results are shown in Figure 9.

According to Figure 8, the strength of the novel naphthol gel remained unchanged after aging for ten days at different salinities. At the salinity of 250,000 mg/L, the gel strength could reach grade G and the dehydration rate was still less than 3%, indicating that the novel naphthol gel had good salt tolerance. The main reasons were as follows: the rigid naphthalene rings and methyl propane sulfonic acid groups contained by the gel system had favorable salt tolerance [15], especially propane sulfonic acid groups, which could effectively resist the invasion of external cations and reduce the gel dehydration; in addition, the structure of three-dimensional network formed after gelatinizing was relatively stable and firm.

With the increase in salinity, the gel dehydration rate increased gradually. By evaluating the gel aged for ten days, the dehydration rate of the gel prepared with 44,000 mg/L water sample was only 0.6%, and that of the gel formulated with 250,000 mg/L water sample was 2.9%. The reasons were as follows: as the salinity increased, the content of cations in the gel system increased, and cations could compress the double electric layers of gel macromolecules; as the aging time prolonged, under the continuous compression of cations, the volume of the gel net shrunk, and water molecules were released from the grid structure to form dehydration; the higher the salinity, the more significant the compression by cations. Moreover, the saline solution competed with the gel system to absorb water molecules by its hydration, further resulting in gel dehydration. The content of Ca^2+^ and Mg^2+^ ions in the simulated formation water from the Tahe River exceeded 10,000 mg/L, and a large number of divalent metal ions was also an important reason for gel dehydration [16]. During gelatinizing, Ca^2+^ and Mg^2+^ ions could bind to carboxyl groups on the macromolecular gel chain, thereby occupying part of the groups and affecting the gel cross-linking; meanwhile, the Ca^2+^ and Mg^2+^ ions in aqueous solution had small radii and relatively strong power to attract water molecules, which also affected the water retention capacity of the gel grid structure.

#### 3.2.3. Gel Viscosity and Gelatinizing Time

To quantitatively and accurately measure the gelatinizing time of naphthol gel, the gel stock solution was prepared with the simulated formation water of the Tahe River and the same batch of gel stock solution was sub-packed into 36 ampoules, which were placed at 120 °C, 130 °C, and 140 °C. One of them was taken out every 2 h. The viscosity of the gelled stock solution and the gel solution at different times and temperatures were separately measured with a viscosimeter and the experimental results are shown in Figure 10.

According to Figure 10, the viscosity of the gel stock solution was 125.3 mPa∙s and the final viscosity of the gelatinized naphthol gel was 42,800 mPa∙s. At 120 °C and within 0–10 h, the gel viscosity barely changed; at 10–20 h, the gel viscosity showed a linear upward change; after 20 h, the viscosity remained unchanged. Thus, the gel was cross-linked from 10 h and wholly gelatinized at 20 h, and the gelatinizing time was 10–20 h. The rule of the gel viscosity changing over time during gelatinizing was similar. The gelatinizing time was 8–18 h at 130 °C and 7–16 h at 140 °C, and it was shortened as the temperature rose. The main reasons were as follows: as the temperature rose, the polymer dynamics in the solution increased, and the polymer macromolecular chains were more prone to entanglement and association to promote cross-linking [17]; in addition, the thermal motions among molecules intensified at high temperatures, and the collision frequency between the cross-linker and amide and carboxyl groups increased, which promoted the occurrence of cross-linking reactions. According to the gelatinizing time at different reservoir temperatures, the degree of gel injection could be adjusted by controlling the injection rate in field construction. The gel stock solution was injected into the target layer before gelatinizing.

## 4. Conclusions

(1)For the high-temperature and high-salinity reservoirs in the Tahe Oil field, AM/AMPS was selected as the polymer, 1,5-dihydroxy naphthol as the cross-linking agent, and polypropylene fiber as the system stabilizer to independently synthesize a novel gel plugging agent, which showed a dense and relatively stable three-dimensional network structure under an SEM.(2)A performance evaluation was conducted on the gel plugging agent and it was found that: as the temperature rose, the gel dehydration rate gradually increased; the high temperature would destroy the gel network structure; the gel was suitable for reservoirs below 140 °C; the gel strength reached grade H after ten days of aging, the dehydration rate was less than 3.6%, indicating good temperature resistance.(3)As the salinity rose, the gel dehydration rate increased gradually; when the gel was aged at 250,000 mg/L salinity for ten days, the dehydration rate was still less than 3%, indicating good salt tolerance.(4)The viscosity of the gel stock solution was mPa∙s and the post-gelatinizing viscosity was 125.3 mPa∙s. The gelatinizing time at 120 °C, 130 °C, and 140 °C was 10–20 h, 8–18 h, and 7–16 h, respectively; as the temperature rose, the gelatinizing time was shortened.

## Figures and Tables

**Figure 1 polymers-14-03894-f001:**
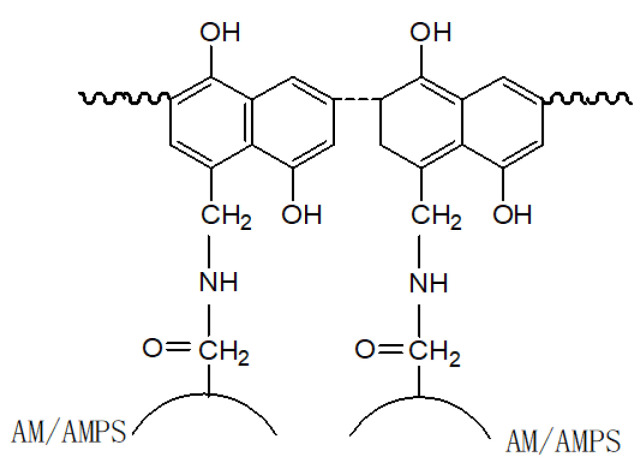
Schematic Diagram of Gel Structure.

**Figure 2 polymers-14-03894-f002:**
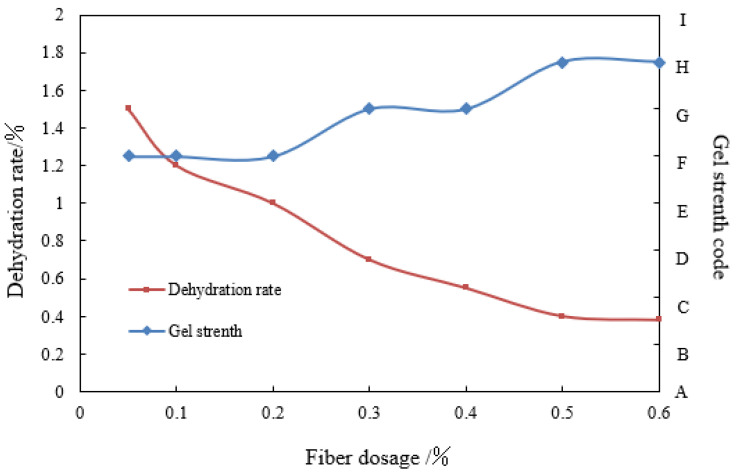
Optimization experiment of stabilizer dosage.

**Figure 3 polymers-14-03894-f003:**
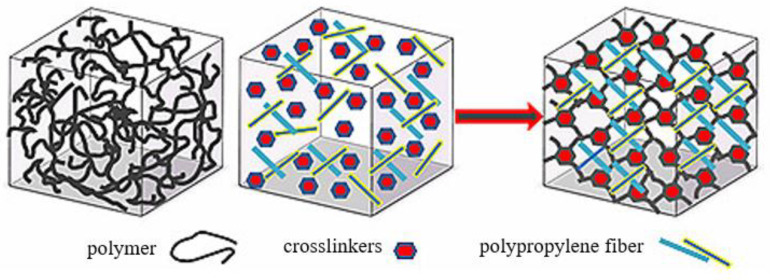
Schematic Diagram of the Skeleton Structure of polypropylene fiber strengthened Naphthol Gel.

**Figure 4 polymers-14-03894-f004:**
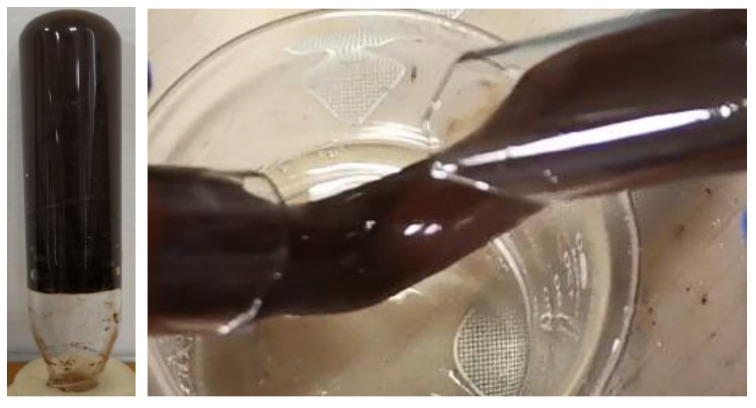
Morphology of Post-gelatinizing Naphthol Gel.

**Figure 5 polymers-14-03894-f005:**
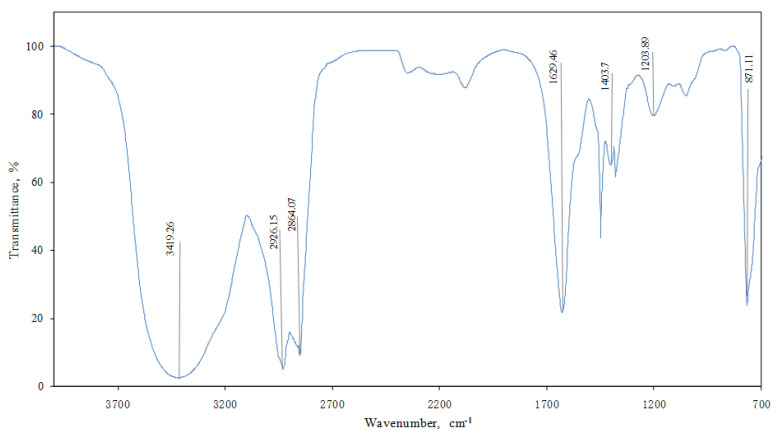
FT-IR Analysis of New Naphthol Gels.

**Figure 6 polymers-14-03894-f006:**
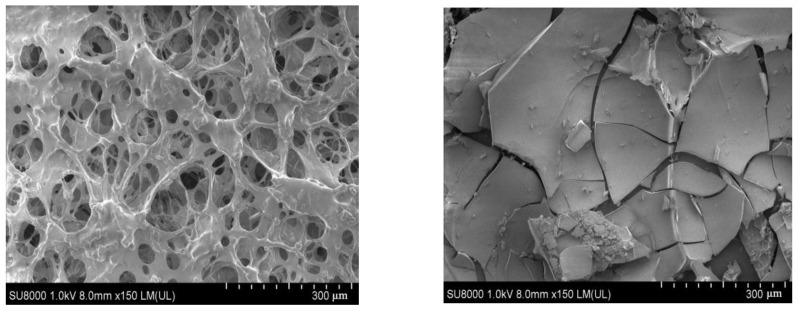
SEM Comparison of Common Chrome Gel (**Left**) and Novel Naphthol Gel (**Right**).

**Figure 7 polymers-14-03894-f007:**
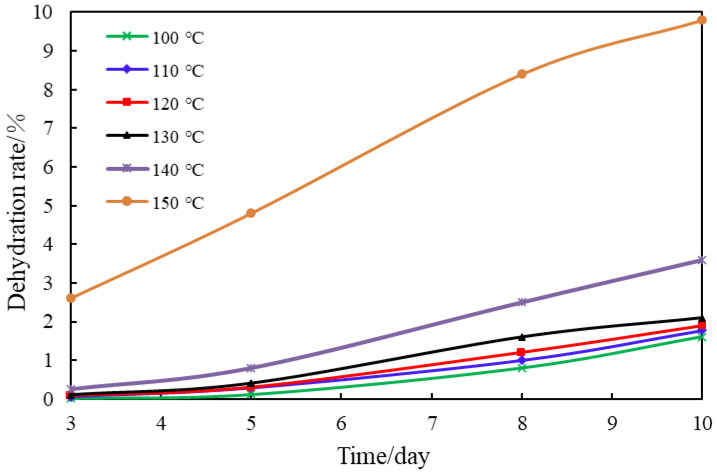
Effect of Temperature on the Gel Dehydration Rate.

**Figure 8 polymers-14-03894-f008:**
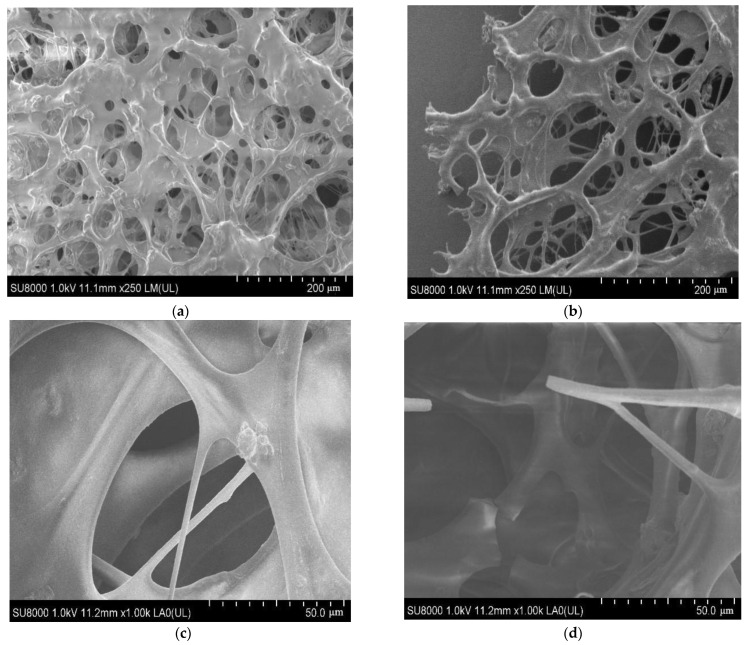
SEM images for the xxx. (**a**) gel structure after aging at 130 °C; (**b**) gel structure after aging at 150 °C; (**c**) gel structure after aging at 130 °C; (**d**) gel structure after aging at 150 °C.

**Figure 9 polymers-14-03894-f009:**
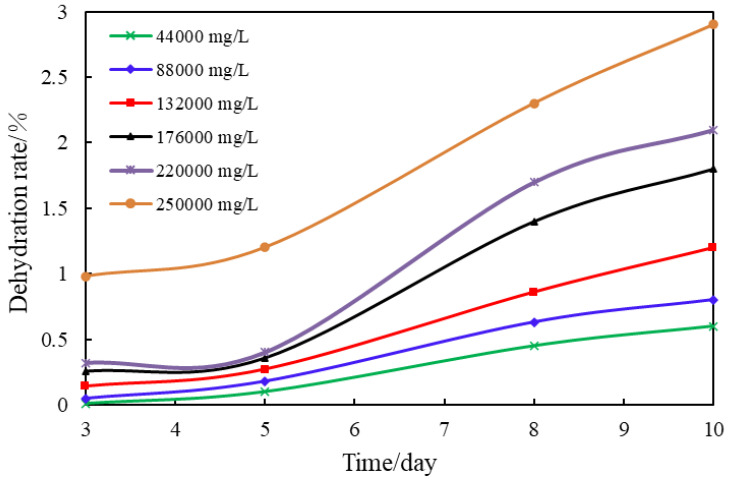
Effects of Salinity on the Gel Dehydration Rate.

**Figure 10 polymers-14-03894-f010:**
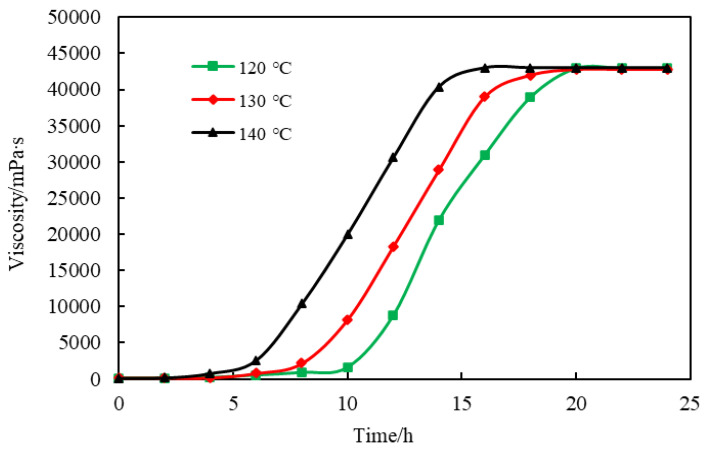
Viscosity–time Curve of Gel.

**Table 1 polymers-14-03894-t001:** Gel Strength Code.

Gel Strength Code	Gel Description
A	No detectable gel formed: the gel appears to have the same viscosity as the original polymer solution
B	Highly flowing gel: the gel seems to be only slightly more viscous than the initial polymer solution
C	Flowing gel: most of the gel flows to the bottle cap by gravity upon inversion
D	Moderately flowing gel: only a tiny portion (5–10%) of the gel does not readily flow to the bottle cap by gravity upon inversion
E	Barely flowing gel: the gel can barely flow to the bottle cap, and a significant portion (>15%) of the gel does not flow by gravity upon inversion
F	Highly deformable non-flowing gel: the gel does not flow to the bottle cap by gravity upon inversion
G	Moderately deformable non-flowing gel: the gel deforms about halfway down the bottle by gravity upon inversion
H	Slightly deformable non-flowing gel: only the gel surface slightly bends by gravity upon inversion
I	Rigid gel: there is no gel surface deformation by gravity upon inversion

**Table 2 polymers-14-03894-t002:** Viscosity Retention of Various Polymers.

Polymer Viscosity/%	6 h	12 h	24 h
AM/AMPS	89	78.4	68.2
AM/DMDAAC	30.2	16.5	7.8
AP915	12	7.5	5.4
HPAM	9.2	5.8	3.5

**Table 3 polymers-14-03894-t003:** Effects of cross-linkers on the Gel Dehydration Rate.

Five Days	Resorcinol	Hydroquinone	Catechol	Polyethyleneimine	Polyethylene Polyamine	A-Naphthol	1,5-Dihydroxy Naphthol
Gel strength	F	F	C	D	C	G	G
Dehydration rate%	9.6	10.8	54.1	32.5	70.3	7.6	6.8

**Table 4 polymers-14-03894-t004:** Effects of Stabilizers on the Gel Dehydration Rate.

30 Days	Polyacrylonitrile Staple Fiber	Lignin Fiber	Polypropylene Fiber	Microsphere Resin	Vitrified Beads	Superfine nylon Six Powder	Rubber Powder
Gel strength	H	H	H	G	D	D	F
Dehydration rate %	6.4	5.3	3	36.5	56.2	50.8	29

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
