# Peer review of "Synthesis and Performance Evaluation of a Novel Heat and Salt-Resistant Gel Plugging Agent"

_polymers, 2022, doi:10.3390/polym14183894_

Round 1
Reviewer 1 Report
This manuscript has interesting result and it has practical value. Please consider the following points to improve the quality of this manuscript:
1. Please briefly describe the used polymers and crosslinkers in introduction.
2. Please briefly describe the method of mixing the polymers and crosslinkers.
3. Please do FT-IR analysis, to justify your explanation in result and discussion section.
Author Response
According to the question you raised, I have carefully revised the original text of the paper one by one. The revised place is yellow and the font is marked in red. please check.

Reviewer 2 Report
The authors reported a study to propose a polymeric system for Tahe Oilfield as gel plugging agent. The authors organized the manuscript in the way to prove their expected outcome. However, there still unclear issues to be clarified; please find below some comments/suggestions which may help to improve the quality of the manuscript:
1. The authors provided the list of materials they have used. However, the source of polymers or the preparation procedures were not mentioned. Therefor the authors are requested to insert description of materials.
2. The authors provided a list of experimental drugs; the reason for presenting this list was not clearly understood as the purpose of the study was to develop plugging agents. The authors are requested to reformulate and insert all details of the experimental procedure so that to be reproducible.
3. The authors mentioned in the title about synthesis of the materials; however, the data presented was rather poor and needs additional information.
Author Response

(The authors gave the same response as above.)

Round 2
Reviewer 1 Report
The manuscript has been revised accordingly, please count the following points
1. ''The pH value of the system is adjusted'' mention the pH
2. Please cite reference in FT-IR analysis
3. Please revise the manuscript by a native speaker
Author Response
- Rather the pH value of the system into 8.
-
2926.15 cm-1 and 2864.07 cm-1 are characteristic peaks of bishydroxynaphthalene ring; 1629.46 cm-1 is the absorption peak of primary amide, and the absorption type is stretching vibration Vibration[11]; 1203.89 cm-1 is the phenolic (C-O) absorption peak (1000-1260), the absorption type is stretching vibration; 871.11 cm-1 is the absorption peak of 1,2,3,4-position four-membered benzene ring substitution, and the absorption type is out-of-plane bending[15].
-
Ask foreign colleagues to help polish the pape.
See the attachment word.

Reviewer 2 Report
The authors answered partially to the addressed issues. More specifically, the query regarding the origin and/or synthesis of polymers was not fully understood. In the last paragraph the authors stated that they have synthesized the acrylamide based polymer and in the experimental section that the polymer was purchased from a company. the authors should state clearly the origin of the polymer and its specifications (molecular weight, other physical chemical characteristics, etc. )
Author Response
The polymer is the finished product purchased from the company.
AM/AMPS was selected as polymer, AM/AMPS polymer was polymerized by acrylamide (AM) monomer and AMPS monomer. All polymers were purchased from Shandong Baomo Biochemical Co., Ltd.
AM/AMPS , with solid content of >88%, AMPS content of >50%, and viscosity average molecular weight of 5 million; AM/DMDAAC、polymer AP915, with viscosity average molecular weight of 5 million and ionic strength of 30%; HPAM, with molecular weight of 8 million and solid content of >88%.